# Nutrition in Cancer Therapy in the Elderly—An Epigenetic Connection?

**DOI:** 10.3390/nu12113366

**Published:** 2020-11-01

**Authors:** Janusz Blasiak, Jan Chojnacki, Elzbieta Pawlowska, Joanna Szczepanska, Cezary Chojnacki

**Affiliations:** 1Department of Molecular Genetics, Faculty of Biology and Environmental Protection, University of Lodz, 90-236 Lodz, Poland; 2Department of Clinical Nutrition and Gastroenterological Diagnostics, Medical University of Lodz, 90-647 Lodz, Poland; jan.chojnacki@umed.lodz.pl (J.C.); cezary.chojnacki@umed.lodz.pl (C.C.); 3Department of Orthodontics, Medical University of Lodz, 92-216 Lodz, Poland; elzbieta.pawlowska@umed.lodz.pl; 4Department of Pediatric Dentistry, Medical University of Lodz, 92-216 Lodz, Poland; joanna.szczepanska@umed.lodz.pl

**Keywords:** cancer, older adults, nutrition, malnutrition, epigenetic regulation of gene expression, DNA methylation, epigenetic diet, caloric restriction

## Abstract

The continuous increase in life expectancy results in a steady increase of cancer risk, which consequently increases the population of older adults with cancer. Older adults have their age-related nutritional needs and often suffer from comorbidities that may affect cancer therapy. They frequently are malnourished and present advanced-stage cancer. Therefore, this group of patients requires a special multidisciplinary approach to optimize their therapy and increase quality of life impaired by aging, cancer, and the side effects of therapy. Evaluation strategies, taking advantage of comprehensive geriatric assessment tools, including the comprehensive geriatric assessment (CGA), can help individualize treatment. As epigenetics, an emerging element of the regulation of gene expression, is involved in both aging and cancer and the epigenetic profile can be modulated by the diet, it seems to be a candidate to assist with planning a nutritional intervention in elderly populations with cancer. In this review, we present problems associated with the diet and nutrition in the elderly undergoing active cancer therapy and provide some information on epigenetic aspects of aging and cancer transformation. Nutritional interventions modulating the epigenetic profile, including caloric restriction and basal diet with modifications (elimination diet, supplementary diet) are discussed as the ways to improve the efficacy of cancer therapy and maintain the quality of life of older adults with cancer.

## 1. Introduction

Aging of societies implies an increasing number of cancer diagnoses in the elderly [1]. As older adults have substantially different nutritional needs than their younger counterparts, the question is whether such differences will result in a different response to cancer therapy in the categories of both the efficacy in the target tissue and unwanted side effects. Any kind of cancer therapy is a serious challenge and burden for the patient, so it should be adjusted to the nutritional status of the patient and vice versa. Nutritional studies among older adults with cancer are considered a major area of interest in geriatric oncology, as most studies on diet and nutrition in cancer have been conducted in younger adults [2].

In general, the care of older adults with cancer is complex due to competing comorbidities, multiple drugs usage, deficit in cognitive functions, and other features complicating the care. On the other hand, cancer chemotherapy may be associated with adverse events, including vomiting and mouth sores, that may influence the nutritional status of cancer patients. Furthermore, cancer is frequently associated with weight loss and a dietary intervention may be recommended in such cases. A European study showed that over 70% of elderly cancer patients presented undernutrition, defined as weight loss of 10% or greater [3].

Epigenetic regulation of gene expression is an emerging field in human molecular genetics, physiology, and pathology. The epigenetic profile of the genome (the epigenome) is established by DNA methylation, chemical modifications of chromatin, and the action of non-coding RNAs. In contrary to its genetic counterpart, the epigenetic profile is erased in the germ cells and can be modulated at any stage of development by environmental and lifestyle influences. This fact is exploited in epigenetic therapies with the use of drugs modulating the epigenetic profile (epidrugs) [4]. Many studies show that diet and nutrition influence epigenetic mechanisms playing a role in the pathogenesis of many diseases, including cancer (reviewed in [5]). On the other hand, the epigenetic profile is modulated by aging. Therefore, epigenetics seems to be a natural candidate to link nutrition with cancer therapy in older adults. In this review, we discuss the main problems associated with nutrition in older adult cancer patients undergoing active therapy, as well as the role of the epigenetic profile in aging and cancer transformation, and present a perspective of epigenetic nutritional intervention in elderly cancer patients.

## 2. Management of Older Adult Cancer Patients

Although chronologic age is one of the main determinants of therapeutic strategy in cancer, older adults have other conditions that may influence morbidity and mortality independently of metrical age (Figure 1) [6].

These conditions include cognitive impairment, delirium, incontinence, malnutrition, falls, gait disorders, pressure ulcers, sleep disorders, sensory deficits, fatigue, dizziness, and others. They are widespread in older adults and may have a major influence on quality of life and disability. Therefore, doctors should have a tool to quickly assess various aspects of elderly patients to develop an optimal therapeutic strategy as well as monitor and evaluate its consequences. They are listed in the comprehensive geriatric assessment (CGA), a process used to evaluate and manage fit, frail, or vulnerable older people (Figure 2) [7]. 

CGA involves not only medical diagnoses but also functional deficiency and the environmental and social matters that disturb patient wellbeing. It creates problem lists and shows aim-driven interventions to face them. Eventually, it delivers and organizes a complex plan for therapy, rehabilitation, support, and long-term care [7].

CGA factors that may be useful in oncology care of older adults are physical function, comorbid medical conditions, cognitive function, psychological state, social support, polypharmacy, and geriatric syndromes [6]. Financial consideration is also included in these factors, often with social support. These factors should be considered in a decision-making process in the treatment of older adults with cancer. Comorbid medical conditions seem to be critical for life expectancy and treatment tolerance, which is essential to maintain quality of life. Moreover, these comorbid conditions usually, if not always, affect the treatment. Therefore, the basic question a doctor should answer is whether a patient is more likely to die of cancer or other comorbid conditions, which is a complex and challenging task in the case of older adults [6]. From the point of view of this review, comorbidity resulting from nutritional status is of a prime interest. However, it is not easy to determine the involvement of dietary factors in the pathogenesis of many serious diseases influencing cancer treatment in the elderly, including other cancers. That is why we will focus on the existing nutritional status of older cancer patients.

In general, weight loss in late life was associated with an increased mortality [8]. Malnutrition in older adults with cancer may diminish tolerance to therapy and result in a worse response to treatment [9]. The risk associated with nutritional status in the senior population can be quickly evaluated with the Mini-Nutritional Assessment (MNA), a part of CGA, including anthropometric measurements; questions related to lifestyle, mobility, and medications; a brief dietary questionnaire; and self-perception of health and nutrition [10]. It can be an alternative or supplement for self-reported practical markers of frailty, including weight loss and low Body Mass Index (BMI), which was established as less than 18.5 kg/m^2^ by the World Health Organization [11]. 

In a multicenter study, Soubeyran et al. enrolled over 300 patients older than 70 years with various types of advanced cancer [12]. They evaluated their state with various aspects of baseline abbreviated CGA and concluded that a low MNA score and poor mobility predicted an early death—within 6 months from the start of chemotherapy. These studies confirm that a poor nutritional status in older adults with cancer is correlated with a bad prognosis. The authors underlined that the MNA test in these patients likely reflected the consequences of advanced disease and that the MNA questionnaire contained 18 questions not related directly with nutrition. Yet another method for comprehensive nutritional assessment of adult oncology patients to determine the strategy of nutritional intervention is SGA (Subjective Global Assessment) and its variant, PG-SGA (Patient Generated-Subjective Global Assessment) [13].

Aaldriks et al. enrolled 143 patients aged 70 years or older with advanced colorectal cancer receiving adjuvant or palliative chemotherapy [14]. Before chemotherapy, they were assessed by MNA, Informant Questionnaire on Cognitive Decline in the Elderly (IQCODE), Groningen Frailty Indicator (GFI), and Mini Mental State Examination (MMSE). The authors observed that malnutrition and frailty were strongly linked with an increased mortality risk in patients undergoing palliative chemotherapy and a poor score on MNA was correlated with a worse tolerance of chemotherapy. Therefore, nutrition was again shown to be an important factor in the cancer care of older adults. 

Comorbidity is one of the most important issues addressed in geriatric assessment. As older age is associated with frailty, diabetes, and cancer, Liuu et al. investigated older adults with cancer from the prospective single-center cohort ANCRAGE (Analyses of CanceR in AGEd) in order to determine the influence of type 2 diabetes mellitus (T2DM) and its vascular complications on frailty and adverse outcomes during 8-year follow-up [15]. They recruited nearly 1100 patients ≥ 75 years with cancer, and about 30% of them presented a metastatic disease, and frailty was common in this group (84%). After adjustment for age, gender, and metastatic status, frail T2DM patients with vascular complications displayed the highest risk of all-cause death. In the context of this review, the most important result of this study was that death was more often due to non-cancer causes, which supports the complexity of considerations surrounding the care of older adults with cancer. On the other hand, it is not easy to assess the real role of cancer in deaths whose immediate reason was T2DM, as cancer and T2DM have much in common and affect each other [16]. 

## 3. Nutrition, Aging, and Cancer in the Elderly

Older adults show diminished energetic demands, but they still need some essential nutrients, which are especially important as their total intake of food is lower than average. Therefore, their diet should be carefully chosen with limited amounts of products with sugar and fat and a dominating proportion of products with high nutrient density. However, cancer as a systemic disease may enforce alterations to such carefully established diets, and cancer therapy may require further changes. Despite common use of dietary supplements after cancer diagnosis, no consensus has been achieved for their recommendation by medical authorities, including the World Cancer Research Fund and the American Cancer Society [2].

Many dietary supplements administrated to cancer patients contain antioxidants that may neutralize reactive oxygen species (ROS) that play a role in the process of carcinogenesis, as they may induce mutations fueling cancer transformation [17,18]. However, many regimes of chemotherapy and radiotherapy produce ROS that can damage biological molecules, including proteins and DNA, in cancer cells. Supplementary antioxidants add to the cellular antioxidant defense system, containing antioxidant enzymes, DNA repair, and low-weight antioxidants. Many studies suggest that this system declines with aging [19,20]. At present, clinical recommendations say that cancer patients, independently of age, should rather not take antioxidants during therapy [21,22,23]. In a recent study, Ambrosone et al. concluded that the use of antioxidant supplements during chemotherapy, as well as iron and vitamin B12, might increase the risk of breast cancer recurrence and mortality [24]. However, this study did not stratify patients according to age, and the main age of patients enrolled in the study was about 50 years.

Malnutrition arises from an inflammatory state, which advances anorexia and resulting weight loss. Malnutrition is common in cancer patients, as up to 40% of all cancer patients display weight loss at the time of diagnosis [25,26]. However, older adults may show weight loss as a result of various comorbidities and other geriatric syndromes, so it is not easy to precisely determine cachexia among them. On the other hand, obesity, the other face of malnutrition, is increasingly becoming an issue affecting cancer survival [27,28]. This problem may be especially important in older cancer patients, as obesity occurs with aging, despite a reduction in food consumption (reviewed in [29]). Weight gain and obesity among older adults may occur with concomitant reduction in muscle mass and sarcopenia [30]. However, steroids and hormonal therapy in a long-term cancer treatment may stimulate the development of diabetes and cardiovascular disease at which older adults, especially with obesity, are at risk [31,32]. Therefore, nutritional research is needed among obese older cancer patients to establish prognosis of the disease course [33].

## 4. Nutrition and Cancer Therapy in the Elderly 

Nutrition care during active cancer therapy should be directed to increase the efficacy of the therapy, reduce unwanted side effects, prevent nutritional deficiencies, and maintain weight and quality of life [34]. Nutritional status is an independent predictor of survival, and poor nutritional status is associated with worse outcomes for older patients undergoing cancer therapy [12,35,36]. On the other hand, malnutrition may be a risk factor for unwanted side effects of chemotherapy [37,38]. 

Chemotherapy influences patients’ nutritional status, as more than half of patients undergoing chemotherapy experience vomiting, mucositis, nausea, and parageusia [39]. Similar effects can be expected in a substantial proportion of cancer patients undergoing radiotherapy [40]. Consequently, malnutrition is an important element that should be considered in the planning of and during cancer therapy. Optimally, malnutrition should be recognized prior to surgery, chemotherapy, and radiotherapy, or any other therapy, and treated with a nutritional intervention [41]. Therefore, nutritional interventions should be fundamental and adjuvant for any kind of cancer therapy as a kind of multidisciplinary follow-up [9]. When patients are of an advanced age, this issue becomes more complex and requires some additional and specific approaches. 

Muscle mass loss and fatty muscle infiltration are frequently used to assess malnutrition, sarcopenia, and cachexia and to monitor the side effects of cancer therapy [42]. Cancer-independent, significant muscle mass loss in older adults is an important factor that should be considered in such assessments. 

Apart from problems associated with cancer therapy and directly related to the diet and nutrition status, some other factors should be considered in the cancer care of older patients. Hoppe et al. presented data from 12 centers in France with older (age ≥70 years) cancer patients receiving first-line chemotherapy [43]. They observed that a substantial portion of patients, 50 of 364, experienced an early functional decline between the beginning of chemotherapy and its second cycle. This decline was determined as a decrease of ≥0.5 points on the Activities of Daily Living scale [44]. Factors associated with early functional decline were evaluated with the use of various geriatric assessments, including abbreviated CGA, MNA. They observed that early functional decline resulting from first-line chemotherapy was associated with baseline depression and instrumental dependencies. Both these features may cause nutritional problems and impede nutritional interventions. 

The diet seems to be the only element during cancer therapy that can be perceived by a patient as a fully controllable means to maintain energy and activity and successfully overcome the therapy [9]. This seems especially important in the case of head and neck cancer as well as cancer of the gastrointestinal tract, as patients with these cancers are particularly prone to problems with nutrition due to the location of tumor and area of treatment [45,46].

From the clinical point of view, future research should concentrate on energy balance among older adults and their body composition during cancer treatment, biomarkers for cachexia, and personalized multi-disciplinary interventions [47]. From a scientific standpoint, it is important to determine the process of cellular aging in cancer cells and relate it to organismal aging. 

## 5. Epigenetic Mechanisms in Cancer Transformation and Aging

Cancer, a disease of genes, results from the accumulation of genetic and epigenetic alterations and their clonal expansion in proliferating cells (Figure 3).

Cancer is predominantly a disease of later life, as it needs time to disrupt controls in multiple cells. Age is frequently considered to be the most serious cancer risk factor, but it is difficult to fully accept this view, especially in cancers underlined by germ mutations or some juvenile leukemias [1]. Genomic and epigenomic instability seem to be crucial for cancer development. Epigenetic dysregulation plays a role in all stages of cancer transformation. It can be induced by genetic changes, first mutations in genes encoding epigenetic regulators, or by tissue inflammation affecting cell signaling, resulting in altered chromatin organization [48].

Dysregulated DNA methylation is likely the best-known epigenetic effect in cancer transformation [49]. Loss of DNA methylation at some specific repetitive elements and regulatory sites has been associated with increased genomic instability and chromosomal aberrations, resulting in fusion genes often encoding oncoproteins [50,51]. 

Modulation of chromatin structure through covalent modification of histone N-terminal tails is an essential way to change DNA accessibility during its transcription, replication, damage repair, and a series of other cellular processes [52,53]. The biological outcome of histone modifications is expressed either by a direct modulation of nucleosomal structure or by recruiting downstream proteins that play a role of ‘reader’ or ‘effector’. The histone code is read to recruit proteins that can alter the chromatin structure. Many enzymes involved in establishing the code can contribute to cancer transformation when their activity is aberrant [54].

The role of non-coding RNAs in cancer is an emerging area of research [55].

Genetic, epigenetic, and environmental events driving the process of aging are mutually coupled, as environmental factors, such as smoking, are associated with the production of molecules that may damage DNA and induce mutations (Figure 4). On the other hand, mutations induced by products of normal cellular metabolism may affect the expression of genes responsible for the detoxification of environmental DNA-damaging agents. Furthermore, there is a mutual dependence between aging and genetic, epigenetic, and environmental factors that promote aging. This dependence is a kind of vicious cycle, as the declines in some functions linked with aging may result in an enhanced susceptibility to environmental factors that in turn may result in further declines in these age-related functions. 

It became evident that the extent of DNA methylation decreases with aging and that alterations might induce the abnormal expression of genes important for the aging process [56,57]. Early interest in the association of aging and loss of 5-metC, an indicator of DNA methylation, focused mainly on hypomethylation of some genes important in aging, similarly to tumorigenesis [58]. However, it was shown later that aging mice transcriptionally activated alleles that were epigenetically silenced in their younger age [59]. Issa et al. were the first to observe the age-associated hypermethylation of CpG (cytosine-guanine dinucleotide) islands in age-related genes in different human tissues [60,61,62,63,64]. Several mechanisms may be responsible for increased methylation in CpG islands in the promoters of aging-related genes (reviewed in [65]). 

DNA methylation, a primary epigenetic event, is characterized by a high inter-individual variability that is underlined by different environmental and lifestyle factors, including the diet (reviewed in [66]). However, it is mostly unknown how dietary compounds affect the DNA methylation pattern. The only exception is the tea polyphenol, epigallocatechin-3-gallate, which is known to act as a competitive inhibitor located in a pocket in the active center of an enzyme responsible for DNA methylation [67]. Li et al. observed that glucose restriction in cultures of normal human fibroblasts extended their Hayflick limit [68]. This result cannot be directly translated into the extension of the human lifespan, but cellular senescence is considered to be associated with organismal aging (reviewed in [69]). 

The importance of the chromatin structure, determined mainly by the covalent modifications of histones, in the process of aging has been confirmed by studies on two human genetic diseases: Hutchinson–Gilford (HGPS) progeria syndrome and Werner syndrome, which are characterized by premature aging phenotypes with a shortened life span and are accepted models for studying the biology of aging in humans (reviewed in [70,71]). Both syndromes are characterized by molecular changes that can be linked with normal human aging. Epigenetic alterations are detected in both syndromes, especially HGPS. These include alterations in the histone distribution, telomere attrition, and the function and biosynthesis of miRNAs. 

In general, chromatin structure, which carries much of the epigenetic information, is considered a major element in the epigenetics of the aging process (reviewed in [72]). Packaging DNA into highly organized nucleosomal structure allows for a precise regulation of all genomic processes occurring in the nucleus, including DNA replication, transcription, recombination, and DNA repair through a defined access to DNA. In general, aging is believed to be associated with nucleosomal remodeling increasing the susceptibility to persistent DNA damage [73]. 

The third main element of epigenetic regulation, non-coding RNAs, with broad two categories, short non-coding RNAs (sncRNA) and long non-coding RNAs (lncRNAs), is reported to display some disrupted functions with aging [74,75,76,77]. Mainly, micro RNAs (miRNA) and lncRNAs were studied for their age-related aspects and, in fact, the majority of miRNAs were shown to be downregulated with age [78,79,80]. 

Although far from the main subject of this review, the honeybee (*Apis mellifera*) offers likely the most convenient example of the effects of diet on lifespan mediated by epigenetic mechanisms [81]. Honeybee larvae are not genetically predetermined to be a queen, but the queen phenotype, with a lifespan up to 20 times longer than a worker, results from the diet containing royal jelly [82]. This effect is mediated by DNA methyltransferase 3 changing the DNA methylation profile.

## 6. Epigenetic Link between Nutrition, Aging, and Cancer 

Caloric restriction (CR), a 30–40% reduction in the caloric intake while maintaining adequate nutrition, and rapamycin are known to extend lifespan. Although the exact mechanism of their action in this effect is not known, the epigenome is considered to be their target, along with genome stability, protein quality control, telomere attrition and function, mitochondrial function, nutrient sensing, cellular senescence, stem cell exhaustion, cellular stress responses, and intercellular communication [83,84,85]. 

McCay et al. were the first to report that a CR diet extended the lifespan of mice [86]. Since then, several works have shown a positive correlation between CR and lifespan in various organisms, including yeast, worms, flies, fish, and primates (reviewed in [87]). Two of these works are worth mentioning, as they reported apparently contrasting results on CR and longevity. Colman et al. showed that CR resulted in the extension of lifespan and a reduction in overall mortality of Rhesus monkeys as compared with controls fed an ad libitum diet [88,89]. On the other hand, Mattison et al. demonstrated that Rhesus monkeys fed with a CR diet did not show any lifespan extension, although these animals displayed a reduction in some age-related diseases, including cancer [90]. Contrasting results obtained in these two studies may be underlined by differences in diet composition, which might differentially affect the epigenetic profile, but also the profile itself could be different in the animals in these two experiments, as they were not conducted in the same environmental conditions. Environment and lifestyle have been shown to have a profound effect on the epigenetic profile [5].

Several mechanisms behind the effect of CR on longevity can be considered. The direct consequence of CR is a reduced energy status in the organism and related decrease in blood glucose, insulin, insulin-like growth factor (IGF-1), growth hormones, and other hormones (reviewed in [91]). Diminished status of cellular energy leads to lower mitochondrial activity and consequently lower aerobic respiration, increased adenosine monophosphate / adenosine triphosphate (AMP/ATP) ratio, and increased nicotinamide adenine dinucleotide (NAD+) levels. Further, two cellular nutrient and energy sensors, adenosine monophosphate kinase (AMPK) and sirtuin 1 deacetylase (SIRT1) are activated [92,93,94]. Activated AMPK induces a series of events resulting in reduced fatty acid synthesis, oxidation, and cholesterol synthesis, but active SIRT1 may increase ketogenesis and lipolysis, and decrease glycolysis. Other proteins can be involved in these processes [95].

The impact of CR on the epigenome was initially associated with an increasing stability of the genome by reduction in the loss of DNA methylation [68]. However, later, the Issa’s lab showed that epigenetic drift, including both gains and losses of DNA methylation at various genome sites, was conserved among species and was correlated with lifespan and CR [96]. Recently, Hernando-Herraez showed that mouse stem cells acquire epigenetic drift by the accumulation of stochastic changes of DNA methylation in the promoters of many genes, which leads to altered transcriptional control and the aging of stem cells [97]. Epigenetic mechanisms of anti-aging effects resulting from CR were then postulated and shown in several works [5,91,94,95,98,99,100].

Some tumors, including brain, head and neck, and lung cancers are glucose dependent, so patients with these tumors may benefit from a diet limiting glucose (e.g., a ketogenic diet), but in general, CR is not documented to have an anticancer effect [101]. CR and ketogenic diet result in increased fatty acid oxidation and acetyl-CoA (acetyl-coenzyme A) production, which, in turn, leads to the enhanced production of β-hydroxybutyrate, which is a source of energy for the brain and an inhibitor of glycolysis [102]. Therefore, a CR diet may increase the antioxidant capacity of normal tissues, but this is not the case in cancer cells [103,104].

There is not a strong rationale for a CR diet in malnourished cancer patients. It is even postulated that a high fat and protein diet better fulfills the nutritional requirements of cancer patients than restrictive diets [105]. For obvious reasons, research performed on obese subjects and experimental animals should not be directly related to cancer patients, especially those with advanced age. On the other hand, elderly cancer patients are often malnourished, and it is rather a risky decision for a doctor to recommend any restrictive diet. Currently, only tumors with a strong dependence on glucose should be considered for such dietary intervention, but each case should be treated individually considering other circumstances, especially those associated with the aging-related features of a patient.

It has been shown that the introduction of certain foods, including grapes (resveratrol), soy (genistein), cruciferous vegetables, and green tea, might have a protective effect against aging and cancer [106]. Moreover, several studies showed that a diet containing these substances (an “epigenetic diet”) reduced the incidence of some diseases and is similar in this regard to a CR diet [107,108]. 

## 7. Summary, Conclusions, and Perspectives

Most cancers occur in older adults, and many factors other than chronologic age determine morbidity and mortality and contribute to the strategies surrounding cancer care. Nutritional studies among older adults with cancer are scarce, but the extension in the life expectancy and new therapeutic strategies imply the need for nutritional support and interventions for this group of patients, as recommended by the American Cancer Society and the National Comprehensive Cancer Network [23,109]. 

Considering research performed so far, any restrictive diet, including a CR diet, is not generally recommended for older adults with cancer. This conclusion does not, however, preclude a beneficial effect of such a diet in cancer prevention. Several nutrients included in a CR diet show epigenetic mechanisms of action, modulating DNA methylation, histone modification, and non-coding RNA functions. However, anticancer-preventive action should be clearly distinguished from beneficial effects in cancer, especially in its advanced form. The mechanism of metastasis, the primary cause of cancer-related death, is poorly known, and it involves different molecular events than cancer initiation, promotion, or even invasion, the initial step of metastasis [110]. This problem is complex, as both aging and cancer significantly affect global gene expression at transcriptome, proteome, and metabolome levels, and it is challenging to predict how these changes would be modulated by nutrition. At present, epigenetics seems to be the most promising link between aging, cancer, and nutrition. 

One important feature of epigenetic modifications is that they may be modulated or even reversed by the diet, which is not the case of genetic alterations, first gene mutations, or chromosomal aberrations [111]. Studies performed so far indicate that not only the kind (quality) of the diet but also the amount of energy (calories) may be important for this modulation. So, what is the kind of diet recommended for older adults undergoing cancer therapy? Of course, it is not easy to give a general answer to this question, as it depends on the cancer type and the kind of therapy. Is a so-called “epigenetic diet” a solution? Does such a diet really exist? Although an “epigenetic diet” is sometimes defined as a diet affecting the epigenome, in fact, a diet that would not affect the epigenetic profile would be very sophisticated, if possible, at all. As we concluded in our previous work, an “epigenetic diet” is a rather misleading term, as it is hardly possible to find a diet that would not affect the epigenome [112]. Instead, three kinds of diet can be considered to amend the needs of elderly patients undergoing cancer therapy. Firstly, there is the basal diet, which is adjusted to the general state of a patient’s health and kind of cancer. Considering the possibility of epigenome modulation, some compounds should be eliminated from this diet (elimination diet) and/or some should be added (supplementary diet). This is the basal diet, which is adjusted to the specificity of this group of patients. Caloric restriction, which is considered to be a direct way to increase lifespan, should not be recommended in general in older adults undergoing cancer therapy, as such groups of patients face age- and cancer-associated anorexia and cathepsia. 

Further research is needed to identify which elements of the diet most effectively decrease morbidity and mortality among older adults with cancer. Molecular studies on changes in the epigenetic profile in these subjects may provide information on the use of drugs modulating that profile, which may contribute to evidence-based practice. No one expects that any kind of diet will result in cancer regression—the goal should be to optimize cancer therapy and to improve quality of life impaired by advanced age, cancer, and cancer therapy. Studies that integrate geriatric and oncology care with nutrition and the modulation of the epigenome seem to be at present a rationale means in which to provide information on appropriate nutritional support nutrition in older adults with cancer.

## Figures and Tables

**Figure 1 nutrients-12-03366-f001:**
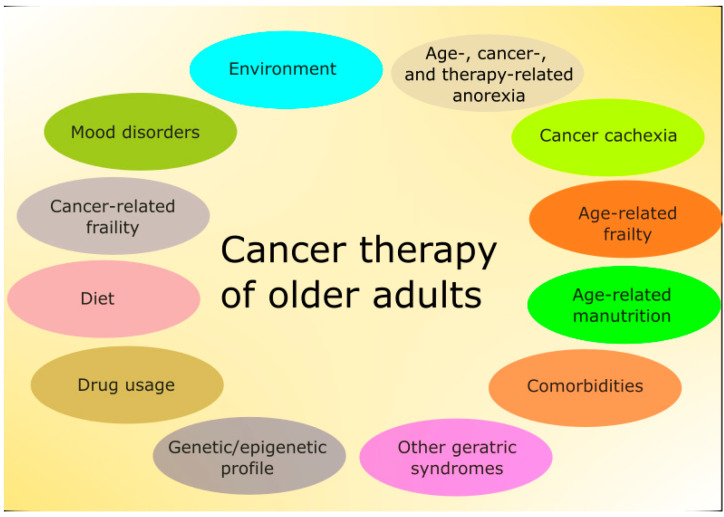
Main factors affecting therapy in older adults with cancer. Some of these factors are mutually dependent and some partly overlap. Environment is understood here in a broad sense and also includes family and social relationships. Some factors, such as the diet, are of general significance, but have several features specific for this group of patients.

**Figure 2 nutrients-12-03366-f002:**
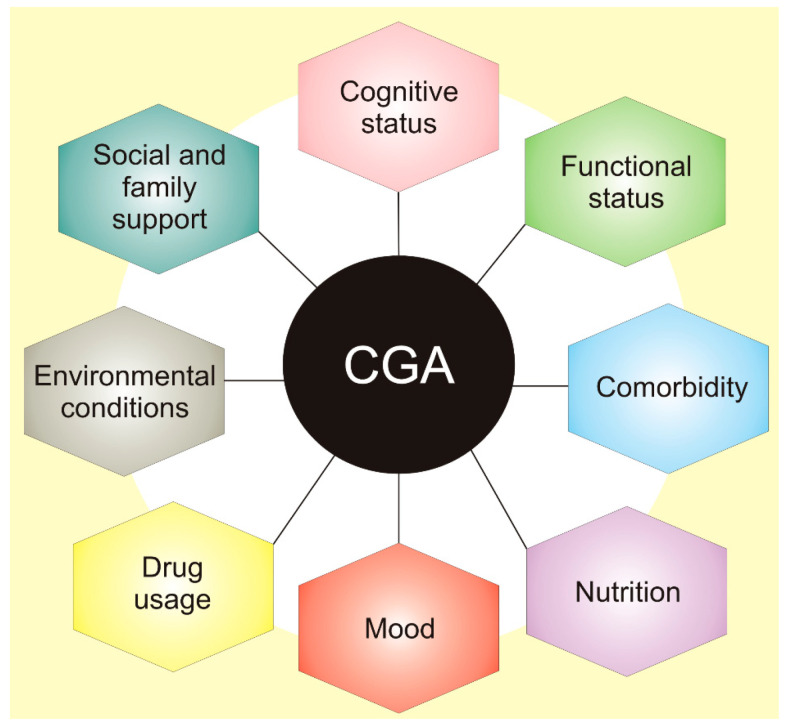
Comprehensive geriatric assessment (CGA) is an organized evaluation method to provide a multidisciplinary assessment of and care for the elderly. It assesses physical medical conditions, including comorbidity, the disease severity, immunization status, and others. Assessment of functional status refers to an elderly person’s ability to perform daily tasks and determines several core functions, including balance and mobility. Other areas of CGA include issues contained in broad categories of assessment of social health and environment.

**Figure 3 nutrients-12-03366-f003:**
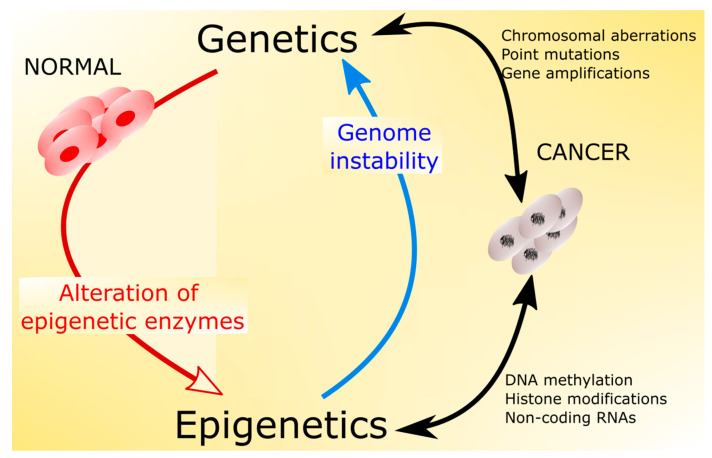
The interplay between genetic and epigenetic factors in cancer transformation. Genomic instability, typical for most cancers, results in an increased number of mutations in genes encoding modifiers of the epigenetic profile. On the other hand, epigenetic changes, including DNA methylation, histone modification, and changes in non-coding RNAs, affect the expression of genes responsible for the maintenance of DNA, resulting in an increased number of chromosomal aberrations, DNA point mutations, amplifications, and other changes increasing genome instability.

**Figure 4 nutrients-12-03366-f004:**
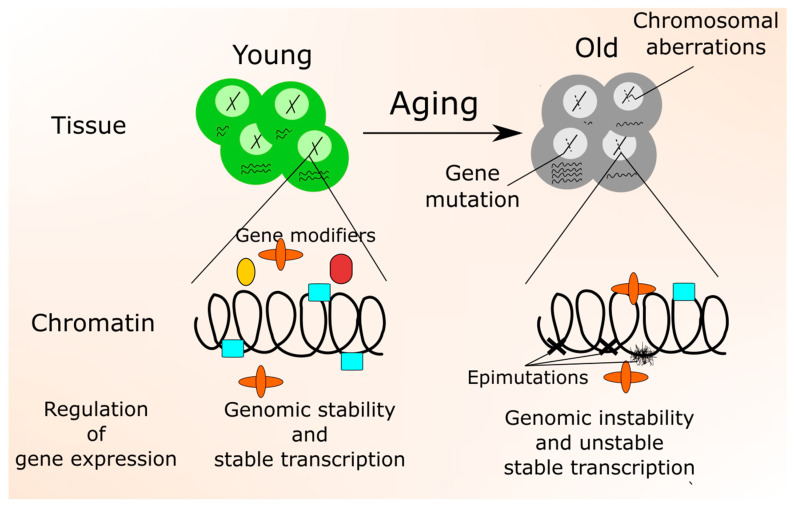
The crosstalk between epigenetic changes, transcription, and genomic instability. In a young organism, transcription is fully controlled and produced the same amount of mRNA in the cells that activate the same genes. In these cells, normal chromatin state and genomic stability are maintained. With increased age, genomic instability increases, resulting in gene mutations and chromosomal aberrations and unstable transcription. Increased DNA damage may result in DNA damage response inducing the recruitment of epigenetic modifiers of chromatin structure and locally resuming its conformation, which may partly stabilize the transcription of neighboring genes. Epimutations, which accumulate in later life, may hamper this process.

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
