# Peer review of "Nutrition in Cancer Therapy in the Elderly—An Epigenetic Connection?"

_nutrients, 2020, doi:10.3390/nu12113366_

Round 1

Reviewer 1 Report

This review focuses on nutrition in older adults with cancer. This group of patients require special, multidisciplinary approach to optimize their therapy and increase their quality of life. In addition, older adults have other conditions that may influence morbidity and mortality independently of chronological age and that may have the major influence on the quality of life and disability. Epigenetic modifications occur with aging and alterations are present in cancer cells. One important feature of epigenetic modifications is that they may be modulated or even reversed by the diet. Parts 5, 6 and 7 of the review are particularly interesting because of the discussion and description of the state of the art on epigenetic mechanims in cancer transformation, with aging, and the epigenetic link between nutrition, aging and cancer. In the conclusion, the authors wonder about the kind of diet recommended for older adults undergoing cancer therapy. They are proposing that further research is needed to identify which elements of the diet most effectively decrease morbidity and mortality among older adults with cancer. According to the authors, nutrition and modulation of the epigenome may be considered to optimize cancer therapy and to improve quality of life in older cancer patients.

The article is well-written, structured, good supported by literature reported in the bibliography. The figures are informative as well. For information, a few typing errors are present in the body text.

I do not have any further comments.

Author Response

Comment: This review focuses on nutrition in older adults with cancer. This group of patients require special, multidisciplinary approach to optimize their therapy and increase their quality of life. In addition, older adults have other conditions that may influence morbidity and mortality independently of chronological age and that may have the major influence on the quality of life and disability. Epigenetic modifications occur with aging and alterations are present in cancer cells. One important feature of epigenetic modifications is that they may be modulated or even reversed by the diet. Parts 5, 6 and 7 of the review are particularly interesting because of the discussion and description of the state of the art on epigenetic mechanims in cancer transformation, with aging, and the epigenetic link between nutrition, aging and cancer. In the conclusion, the authors wonder about the kind of diet recommended for older adults undergoing cancer therapy. They are proposing that further research is needed to identify which elements of the diet most effectively decrease morbidity and mortality among older adults with cancer. According to the authors, nutrition and modulation of the epigenome may be considered to optimize cancer therapy and to improve quality of life in older cancer patients.

The article is well-written, structured, good supported by literature reported in the bibliography. The figures are informative as well. For information, a few typing errors are present in the body text.

I do not have any further comments.

Answer: We have done our best to correct all typos.

Reviewer 2 Report

Thanks for the opportunity to review for nutrients the manuscript: “Nutrition in Cancer therapy in the elderly. An epigenetic connection?” by Blasiak et al. The topic is of interest. Since caloric restriction may increase lifespan, the authors tried to find a possible connection between nutrition, age, and cancer through epigenetics. This is a plausible link.  The idea is fascinating. However, the Authors failed, in my opinion, to argue this thesis.

From a general point of view, the paper seems an opinion paper, not a scientific revision of the literature.  Many references are reported, but there are no new insights for the general readership.

Paragraphs 1-4 are too long and redundant, the text does not provide novel insights for the reader.  Furthermore, some periods are confusing and wrong, others seem only opinions of the Authors, not confirmed by the literature. 

Paragraphs 5-6 are too technical and difficult to understand for an average scholar. The sense of the argument is lost in the technicalities.  

Paragraph 7 is slightly better and fitter than the others for this review.

The conclusion remains too generic.

In details:

ABSTRACT:

I suggest revising the second part of the abstract. It seems like a conclusion of a manuscript rather than an introduction to the topic. Moreover, the abstract should be more focused on the aims of the paper and briefly report the methods of research.

1.INTRODUCTION:

- line 50-51: “Unless it interferes with therapy”... I do not understand what it means: does diet interfere with therapy? Please reformulate.

- line 51: “Displayed” refers to a screen. Please revise the term.

- line 51: malnutrition is a term that needs to be precisely defined.

- line 53: The link between (aging, cancer, nutrition) and (epigenetics) is not immediate.

It should be useful to introduce a definition of “epigenetics” before talking about its importance in cancer and nutrition. I suggest to take 2-3 lines to describe what epigenetics is after lines 53-54. The same for the term “epidrugs”.

- line 60: “the” role

- line 80: “somebodies” (without space)

FIGURE 2:

- line 87-88: “CGA .. It can be used to evaluate elderly patients considering chemotherapy.”  Unless there are strong evidence assessing the prognostic role of CGA in elderly cancer patients, I would not consider it as an assessing tool in this setting.  

  1. Nutrition, aging and cancer in the elderly

- line 148: which “are” (not “is”) especially important

- line 152-153: “but these nutrients may equally feed malignant tumor”: This is quite a confusing and false opinion as highlighted by ESPEN guidelines in nutrition on cancer patients: “Theoretical arguments that nutrients “feed the tumor” are not supported by evidence related to clinical outcome and should not be used to refuse, diminish, or stop feeding”, section A2, statement 11. Arends et al. Clinical Nutrition, 36 (2017). Please remove the period.

- line 153-154: “This may underline different and sometimes conflicting results of trials on the use of high doses of supplements in cancer prevention, cancer therapy and in cancer survivors”. This period is not supported by an evidence-based statement. It is a confounding phrase. Please, remove it.

 Line 156: achievesd

- line 158: “administrated by cancer patients” change in “administered to cancer patients”

- line 163: “antioxidants… they may also decrease the efficacy of the therapy in cancer cells.” If it is not demonstrated by evidence, this is only an hypothesis and should not be reported in a scientific review of the literature. Moreover, it can be confounding for the readers. Please, remove it.

- line 166-167: “older adults may worse protect their normal cells against unwanted effects of ROS-inducing cancer therapy than their younger counterparts” It sounds not fit for a scientific work. Please reformulate.

- line 182: “weight loss in older cancer patients represents cancer cachexia”. It is too simplistic. Cancer cachexia is a multifactorial syndrome that needs to be precisely defined. (Fearon KC, Voss AC, Hustead DS; Cancer Cachexia Study Group. Definition of cancer cachexia: effect of weight loss, reduced food intake, and systemic inflammation on functional status and prognosis. Am J Clin Nutr. 2006 Jun;83(6):1345-50. )

  1. Nutrition and cancer therapy in the elderly

- lines 212-213: “treated with nutritional intervening through nutritional supporting and artificial nutrition”. This period is not clear, please reformulate.

- Line 221-222: “Muscle mass loss and fatty muscle infiltration are frequently used to assess the risk of malnutrition..”. Muscle mass is not used to assess the risk of malnutrition. It is used to assess malnutrition (GLIM Criteria) or sarcopenia. The risk of malnutrition is assessed simply by weight loss, reduction of nutritional intake and BMI (see NRS-2002 or MUST)

- line 246: “important question as to how to provide nutrients for fighting with cancer and not to feed the tumor”. Nutrients do not feed the tumor. See above. The conclusion is confounding.

  1. Epigenetic, mechanisms of aging

This paragraph is difficult to follow. It would be advisable to re-work this part and better link it with the rest of the manuscript.

Line 316 “it is not the aim of this review…”, line 410 “ although far for the main subject of this review..”, the authors themselves realized that periods are off topic. Please remove all information not relevant. (for instance please delete lines 410-414.)

Line 347 Issa “et al.”

  1. Summary, conclusions and perspectives

- line 506: “r”ecommended

Author Response

Comment: Thanks for the opportunity to review for nutrients the manuscript: “Nutrition in Cancer therapy in the elderly. An epigenetic connection?” by Blasiak et al. The topic is of interest. Since caloric restriction may increase lifespan, the authors tried to find a possible connection between nutrition, age, and cancer through epigenetics. This is a plausible link.  The idea is fascinating. However, the Authors failed, in my opinion, to argue this thesis.

From a general point of view, the paper seems an opinion paper, not a scientific revision of the literature.  Many references are reported, but there are no new insights for the general readership.

Answer: We expressed our opinions on the basis of reviewed literature – this is the way we understand “a critical review”. In some places, we presented our vison of further research and that is why our manuscript can be rather classified as a review/perspective and not a “pure review”.

Comment: Paragraphs 1-4 are too long and redundant, the text does not provide novel insights for the reader.  Furthermore, some periods are confusing and wrong, others seem only opinions of the Authors, not confirmed by the literature.

Answer:  Our manuscript is multidisciplinary and link molecular biology with geriatrics and cancer clinic. We realize that majority of readers will be particularly interested in either molecular or clinical aspects. In other words, for clinicians, the clinical aspects will seem to be redundant and molecular aspects will be treated as “technical” and vice versa. We have tried to find the right balance between these two areas of our manuscript. In particular:

Section 1. Introduction

We have removed the fragment (lines 35-39 in the original manuscript):

“Around 54 percent of new cases and 70 percent of mortality from cancer occur in patients ≥ 65 years of age [1]. Despite the fact that cancer mortality rates has been decreasing since 90-ties mainly due to common screening programs and improved diagnostic and therapeutic options, cancer is still a serious medical problem and it is the second leading cause of death worldwide.”

We have changed the sentence:

“As older adults have substantially different nutritional needs than their younger counterparts, the question is whether such difference will result in a different response to cancer therapy, in both the efficacy in the target tissue and unwanted side effects.”

into:

“As older adults have substantially different nutritional needs than their younger counterparts and the question is whether such difference will result in a different response to cancer therapy, in the categories of its efficacy in the target tissue and unwanted side effects.”

We have changed the sentence (last paragraph):

“Epigenetic regulation of gene expression plays and important role in human physiology and pathology and is exploited in emerging epigenetic therapies, in which drug modulating the epigenetic profile, epidrugs, are administrated [4].”

into the following fragment:

“Epigenetic regulation of gene expression is an emerging field in human molecular genetics, physiology and pathology. The epigenetic profile of the genome (the epigenome) is established by DNA methylation, chemical modifications of chromatin and the action of non-coding RNAs. In contrary to its genetic counterpart, the epigenetic profile is erased in the germ cells and can be modulated at any stage of development by environmental and lifestyle influences. This fact is exploited in epigenetic therapies with the use of drugs modulating the epigenetic profile (epidrugs) [4].”

Section 2. Management of older adult cancer patients

We have removed the following sentences/fragments:

“Geriatric syndromes can be identified by a proper assessment.”

“CGA in its simplified form may be carried out by a single person – usually primary care physician or geriatrist, but it can be also performed by a more intense multidisciplinary program. Some bodies prefer to call CGA as comprehensive older age assessment (COAA).”

“There may be a serious discrepancy between results obtained in a group of highly selected individuals and a group of “real-world” patients who may be at increased risk of complications that may be evaluated by CGA or other geriatric assessment.”

We have also made some minor changes to improve the style.

Section 3. Nutrition, aging and cancer in the elderly

We have removed the following fragment:

“Therefore, older adults may worse protect their normal cells against unwanted effects of ROS-inducing cancer therapy than their younger counterparts. However, there is a question about the efficacy of antioxidant defense system in cancer cells. This problem is complex, as it involves a connection between cellular, organismal, and metrical aging, which has not been established yet. It cannot be excluded that cancer cells are entirely new entity in organism and act accordingly to their own program, not necessarily coordinated with the development of the organism. If this is the case, the antioxidant system in cancer cells could be largely independent of the aging of the host organism. In summary – antioxidant supplementation during cancer therapy could decrease the efficacy of the therapy, but protect normal cells against unwanted effects of the therapy and before a recommendation of such supplementation in older adults, studies on the effectiveness of antioxidant defense system in cancer cells in dependence on aging are advised.”

Section 4. Nutrition and cancer therapy in the elderly

We have removed the following fragments from this section:

“Malnutrition may affect treatment results, postpone or even suspend wound healing, worsen muscle function and increase the risk of complications after surgery. All these effects lead to prolonged hospital stay, which may be associated with further complications. Nutritional status and nutrition plan may affect efficacy of anticancer treatment, especially chemo- and radiotherapy. On the other hand, cancer therapy, including surgery and chemotherapy, may have a substantial impact on nutritional status of patients [9].”

 “Weight loss is likely the primary signal for malnutrition in oncology patients, but again advanced age make it complex and this indicator should not be considered singly as a reliable marker for malnutrition. BMI determination should be accomplished by the evaluation of nutrition intake and inflammatory status, which is a standard clinical reference [9,13]. However, this standard recommendation should be supplemented with a panel of issues specific to older age.”

“This is certainly a great challenge, but its addressing in future research may contribute to answer such important question as how to provide nutrients for fighting with cancer and not to feed the tumor or whether cancer cells’ memory include the story of the host organism and in that way cancer cells are balanced with all disadvantages of aging organism. This seems especially important in the context of efficacy of cancer therapy and its modulation by dietary supplements. Are the cancer cells older in older adults or are they brand new as in younger persons?”

Comment: Paragraphs 5-6 are too technical and difficult to understand for an average scholar. The sense of the argument is lost in the technicalities.

Answer: We have combined sections 5 and 6 into one section entitled: 5. Epigenetic mechanisms in cancer transformation and aging, removed large fragments from those sections and simplified the narration. 

Comment: Paragraph 7 is slightly better and fitter than the others for this review.

The conclusion remains too generic.

Answer: We share this opinion, but we do not find a rationale for a more detailed conclusion, e.g. recommendation of any specific kind of the diet.

In details:

ABSTRACT:

Comment: I suggest revising the second part of the abstract. It seems like a conclusion of a manuscript rather than an introduction to the topic. Moreover, the abstract should be more focused on the aims of the paper and briefly report the methods of research.

Answer: We have changed the fragment:

“Caloric restriction (CR) and intermediate fasting are reported to have anti-aging effects that may be partly underlined by epigenetic mechanisms. However, there is not a rationale to apply CR in malnourished patients and it should not be generally recommended for older adults with cancer. Supplements of diet with some chemicals with verified epigenetic effects, including epidrugs, may be considered to improve cancer therapy and decrease its side effects in older adults.”

into:

“In this review, we present problems associated with the diet and nutrition in the elderly undergoing active cancer therapy and provide some information on epigenetic aspect of aging and cancer transformation. Nutritional interventions modulating the epigenetic profile, including caloric restriction and basal diet with modifications (elimination diet, supplementary diet) are discussed as the ways to improve efficacy of cancer therapy and maintain quality of life of older adults with cancer.”

1.INTRODUCTION:

Comment: - line 50-51: “Unless it interferes with therapy”... I do not understand what it means: does diet interfere with therapy? Please reformulate.

Answer: We have changed the sentence:

“Furthermore, cancer itself is frequently associated with weight loss, so the diet is needed to gain weight unless it interferes with the therapy.”

into:

“Furthermore, cancer is frequently associated with weight loss and so a dietary intervention may be recommended in such cases.”

Comment: - line 51: “Displayed” refers to a screen. Please revise the term.

Answer: We have changed the sentence:

“A European study showed that over 70% of cancer patients displayed malnutrition, defined as weight loss of 10% or greater [3].”

into:

“A European study showed that over 70% of elderly cancer patients presented undernutrition, defined as weight loss of 10% or greater [3].

Comment: - line 51: malnutrition is a term that needs to be precisely defined.

Answer: We have changed “malnutrition” into “undernutrition”. We wrote on the other aspect of malnutrition, obesity, in the 3. section.

Comment: - line 53: The link between (aging, cancer, nutrition) and (epigenetics) is not immediate.

It should be useful to introduce a definition of “epigenetics” before talking about its importance in cancer and nutrition. I suggest to take 2-3 lines to describe what epigenetics is after lines 53-54. The same for the term “epidrugs”.

Answer: We have added the following fragment to the last paragraph in Introduction section:

“Epigenetic regulation of gene expression is an emerging field in human molecular genetics, physiology and pathology. The epigenetic profile of the genome (the epigenome) is established by DNA methylation, chemical modifications of chromatin and the action of non-coding RNAs. In contrary to its genetic counterpart, the epigenetic profile is erased in the germ cells and can be modulated at any stage of development by environmental and lifestyle influences. This fact is exploited in epigenetic therapies with the use of drugs modulating the epigenetic profile (epidrugs) [4].”

Comment: - line 60: “the” role

Answer: Done.

Comment: - line 80: “somebodies” (without space)

Answer: That fragment has been removed in the revision.

Comment: FIGURE 2:- line 87-88: “CGA .. It can be used to evaluate elderly patients considering chemotherapy.”  Unless there are strong evidence assessing the prognostic role of CGA in elderly cancer patients, I would not consider it as an assessing tool in this setting. 

Answer: We have removed that sentence.

Comment: Nutrition, aging and cancer in the elderly

- line 148: which “are” (not “is”) especially important

Answer: Done

Comment: - line 152-153: “but these nutrients may equally feed malignant tumor”: This is quite a confusing and false opinion as highlighted by ESPEN guidelines in nutrition on cancer patients: “Theoretical arguments that nutrients “feed the tumor” are not supported by evidence related to clinical outcome and should not be used to refuse, diminish, or stop feeding”, section A2, statement 11. Arends et al. Clinical Nutrition, 36 (2017). Please remove the period.

Answer: We have removed that sentence

Comment: - line 153-154: “This may underline different and sometimes conflicting results of trials on the use of high doses of supplements in cancer prevention, cancer therapy and in cancer survivors”. This period is not supported by an evidence-based statement. It is a confounding phrase. Please, remove it.

Answer: We have removed that sentence

Comment: Line 156: achievesd, - line 158: “administrated by cancer patients” change in “administered to cancer patients”

Answer: Done.

Comment: - line 163: “antioxidants… they may also decrease the efficacy of the therapy in cancer cells.” If it is not demonstrated by evidence, this is only an hypothesis and should not be reported in a scientific review of the literature. Moreover, it can be confounding for the readers. Please, remove it.

Answer: We have done so.

Comment: - line 166-167: “older adults may worse protect their normal cells against unwanted effects of ROS-inducing cancer therapy than their younger counterparts” It sounds not fit for a scientific work. Please reformulate.

Answer: That was included in the fragment, which has been removed in the revision.

Comment: - line 182: “weight loss in older cancer patients represents cancer cachexia”. It is too simplistic. Cancer cachexia is a multifactorial syndrome that needs to be precisely defined. (Fearon KC, Voss AC, Hustead DS; Cancer Cachexia Study Group. Definition of cancer cachexia: effect of weight loss, reduced food intake, and systemic inflammation on functional status and prognosis. Am J Clin Nutr. 2006 Jun;83(6):1345-50. )

Answer: We have removed that sentence.

Comment: Nutrition and cancer therapy in the elderly

- lines 212-213: “treated with nutritional intervening through nutritional supporting and artificial nutrition”. This period is not clear, please reformulate.

Answer:  We have removed the end part of that sentence (through nutritional supporting and artificial nutrition).

Comment: - Line 221-222: “Muscle mass loss and fatty muscle infiltration are frequently used to assess the risk of malnutrition..”. Muscle mass is not used to assess the risk of malnutrition. It is used to assess malnutrition (GLIM Criteria) or sarcopenia. The risk of malnutrition is assessed simply by weight loss, reduction of nutritional intake and BMI (see NRS-2002 or MUST)

Answer: we have changed “to assess the risk of malnutrition” into “to assess malnutrition”

 Comment: - line 246: “important question as to how to provide nutrients for fighting with cancer and not to feed the tumor”. Nutrients do not feed the tumor. See above. The conclusion is confounding.

Answer: This expression was in a fragment that has been removed in the revision.

Comment:    Epigenetic, mechanisms of aging

This paragraph is difficult to follow. It would be advisable to re-work this part and better link it with the rest of the manuscript.

Line 316 “it is not the aim of this review…”, line 410 “ although far for the main subject of this review..”, the authors themselves realized that periods are off topic. Please remove all information not relevant. (for instance please delete lines 410-414.)

Answer: We have made substantial changes to these sections, which are outlined in answers to general comments.

Comment: Line 347 Issa “et al.”

Answer: Done.

Comment: Summary, conclusions and perspectives

- line 506: “r”ecommended

Answer: Done.

Reviewer 3 Report

This is an interesting review that discussed the epigenetic connection between nutrition and cancer therapy in the elderly. In general, the manuscript is well written; please revise the English language and check throughout the text for spelling errors. Perhaps, in the Abstract the aim of the review should be stated more clearly. Figures are detailed and helpful for the reader. A summative table of the studies on the topic should be added. I would suggest to include further discussion on the role of the Mediterranean diet against cancer due to the intake of different substances as demonstrated in a number of studies (i.e. Borzì AM et al,  Nutrients. 2018; Grosso G et al, Nutr Cancer. 2014).

Author Response

Comment: This is an interesting review that discussed the epigenetic connection between nutrition and cancer therapy in the elderly. In general, the manuscript is well written; please revise the English language and check throughout the text for spelling errors.

Answer: We have done our best to correct all typos.

Comment: Perhaps, in the Abstract the aim of the review should be stated more clearly.

Answer: We have changed the fragment:

“Caloric restriction (CR) and intermediate fasting are reported to have anti-aging effects that may be partly underlined by epigenetic mechanisms. However, there is not a rationale to apply CR in malnourished patients and it should not be generally recommended for older adults with cancer. Supplements of diet with some chemicals with verified epigenetic effects, including epidrugs, may be considered to improve cancer therapy and decrease its side effects in older adults.”

into:

“In this review, we present problems associated with the diet and nutrition in the elderly undergoing active cancer therapy and provide some information on epigenetic aspect of aging and cancer transformation. Nutritional interventions modulating the epigenetic profile, including caloric restriction and basal diet with modifications (elimination diet, supplementary diet) are discussed as the ways to improve efficacy of cancer therapy and maintain quality of life of older adults with cancer.”

Comment: Figures are detailed and helpful for the reader. A summative table of the studies on the topic should be added.

Answer: There are too few studies on the topic of the review – in fact there is not a study showing effect of epigenetic dietary intervention in older adults with cancer.

Comment: I would suggest to include further discussion on the role of the Mediterranean diet against cancer due to the intake of different substances as demonstrated in a number of studies (i.e. Borzì AM et al,  Nutrients. 2018; Grosso G et al, Nutr Cancer. 2014).

Answer: The Mediterranean diet is well known for its beneficial effects in cancer prevention, but there is no evidence that its effects may be different in older adults with cancer than their younger counterparts. That is why we did not include the Mediterranean diet in our discussion, similarly to several other “healthy” diets.

Round 2

Reviewer 2 Report

The authors made significant changes. The manuscript has significantly improved. 

I have no further comments.